# The Salt without the Girl: Negotiating Embodied Identity as an Agender Person with Cystic Fibrosis

**Alexandra C.H. Nowakowski** 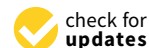

College of Medicine, Florida State University, Tallahassee, FL 32306, USA; xnowakowski@fsu.edu

**Abstract:** In this manuscript, I build and expand on prior work by myself (Nowakowski, 2016) and others exploring the dynamics of embodiment among people with chronic health conditions. Specifically, I critically investigate the intersecting social and medical elements of responses to bodies perceived as too thin and otherwise lacking in physical ability, using my own experiences of living and aging with cystic fibrosis (CF) as a case example. In these explorations, I center gendered identity and its intersection with disabling physical illness. I do so by using my own lived experiences as autoethnographic anchor points to guide critical review of key concepts from the nexus of these two content areas. I focus throughout on exploring how others' reactions to a frail-looking body often constitute a form of forced gendering via the narratives people attempt to construct for why a person's body appears that way. The title of the manuscript supports this framework by referencing three cornerstones of patient experience in the CF community: the general trend of patients having salty skin due to the pathology of the disease, a prior embodiment project called Salty Girls (Pettigrew, 2012) that engages this idea, and the more abstract concept of "saltiness" in describing the grit marginalized people display in responding to microaggressions.

**Keywords:** gender; nonbinary; cystic fibrosis; embodiment; agender identities; identity negotiation

## 1. Introduction

As a medical sociologist, I spend a substantial amount of time studying the complexities of people's bodies, identities, and lives. Yet I can convey a plethora of important information about my own experience with two simple truths: I am agender, and I have cystic fibrosis. In this narrative, I will explore the intersection of these two defining elements of my personal biography in the context of embodiment, beginning with an explanation of what each term means.

Being agender means not having an intrinsic gender identity (Bosse and Chiodo 2016). The specific ways in which people who identify as agender experience this phenomenon vary (Nicolazzo 2016). Yet being agender is generally understood as not seeing ourselves through a gendered lens, even though we may be very aware of the gendered ways in which others view us (Galupo et al. 2017). Agender identity is one of many ways of experiencing oneself outside of binary concepts of gender (Nicolazzo 2016). It is thus part of the general group of identities that "queer" gender (Richards et al. 2016). Some agender individuals may thus identify as genderqueer in certain social and political contexts (Nicolazzo 2016). Genderqueer populations may also include, but are not limited to, people with the following identities: genderfluid, androgynous, or pangender (Galupo et al. 2014).

What sets agender identity apart from others in the genderqueer demographic is the lack of a gender concept within oneself (Bosse and Chiodo 2016). Because we have no intrinsic gender identity at all as opposed to a specific gendered identity that is viewed as non-normative, we often experience our nonbinary selves differently than other genderqueer people do (Weisberg and Tompkins 2015). Relative to other nonbinary people, agender individuals often experience both less dysphoria (a feeling of not being fully oneself physically) and less stigma (a feeling of being ostracized by others socially)

(Galupo et al. 2014). We also tend to find "stealth" more easily within our reach, possibly because we often do not present our gender in any highly specific manner (Nicolazzo 2016).

Cystic fibrosis (CF) is a complex and progressive chronic disease. It is genetic; most people with the disease experience symptoms beginning at birth (Cutting 2015). It is also associated with earlier death, despite recent advances in treatment that have prolonged life expectancy (Nowakowski and Sumerau 2019). The disease results from problems with a protein that moves chloride (a salt component) in and out of cells. People have certain body parts that lubricate themselves; these are called mucous membranes. In people without CF, the presence and work of this specific protein helps the mucous membranes produce a thin and slippery substance that works very well as a lubricant. This substance helps things move around. By contrast, in people with CF, the mucous membranes instead produce something akin to rubber cement. This substance is not merely a poor lubricant, but rather an outright hindrance to movement (Lyczak et al. 2002). As a result, small parts of our bodies that need to move regularly, such as the tiny hairs (cilia) that clean our mucous membranes, cannot do their jobs (Gibson et al. 2003).

People with CF wind up with a variety of health problems because of this basic process. These include chronic infections in our lungs and other organs, blockages of small tubes in organs like the kidneys and liver, low release of substances like digestive enzymes and insulin from our pancreases, alternating bouts of constipation and diarrhea, and a host of damages to other parts of our bodies that result from these issues in the mucous membranes (Emerson et al. 2002). People often think about CF as a lung disease, but this is a misconception. CF is a whole-body disease, one that impacts most everything else in the lives of patients (Segal 2008). Its specific impacts also vary across patients, in part because many systems in the body are affected and in part because there are over 2000 known genetic mutations that can produce the disease (Cutting 2015).

Perhaps unsurprisingly, both people living with CF and people identifying as agender often find that we have a lot in common with one another based on just those single shared characteristics. Yet in both cases, we are also diverse in a number of important ways. Indeed, one of the most visible representations of people with CF in public consciousness is a photography project called *Salty Girls* that speaks deeply to this diversity. After being diagnosed with CF himself, photographer Ian Ross Pettigrew initiated a portfolio of work profiling others with the disease. He soon noticed that many of his subjects looked female, and began focusing in on this group of fellow CF community members. The result was the *Salty Girls* project, which is now in its second phase.

The *Salty Girls* project illustrates a variety of dynamics in the gendering of female-looking bodies with CF, both by the photo subjects themselves and by other people. Describing the participants as "girls" may not be substantively appropriate for each person profiled. Indeed, the gender presentation and expression of different people in the original *Salty Girls* photostream varies substantially. This can be construed as both a strength and a weakness of the project; a strength because of the visibility and empowerment that depicting diverse bodies and genders among female-looking people with CF can foster, and a weakness because of the conflation of looking female with being a girl. Pettigrew's own blog writings about his photography, including both *Salty Girls* and other projects, suggest that he is both aware of these dynamics and interested in exploring them through art. The use of the term "girl" to refer to female-looking adults with CF also introduces an additional sociological dynamic (Duncan and Messner 1998). It speaks to the general infantilization of female-looking bodies that intensifies in cases of chronic illness (Nowakowski 2016). It also evokes the dualistic relationships that people with CF often have with the aging process, a concept I explore later in this manuscript.

Predictably, a lot of the bodies depicted in *Salty Girls* are very thin. Pettigrew does not select participants based on weight; indeed, a range of body sizes, shapes, and compositions appear in the project photostream. However, the overall trend favors thinness and also intersectionally privileged characteristics such as whiteness and higher economic class. Interactionist sociology explores how people's appearance, posture, clothing, and accessories communicate information about their social status (Schwalbe and Shay 2014). In all cases, these characteristics indicating relatively

higher socioeconomic status are features of the demographics of CF in adults (McCormick et al. 2010). People with CF are more likely to look white because of the higher incidence of the disease in European-descended populations (Hamosh et al. 1998). We are also more likely to be thin because of the high prevalence of exocrine pancreatic insufficiency in adults with the disease (Sawicki et al. 2009). Furthermore, people with CF whose families have higher levels of income and wealth are more likely to survive into adulthood because of their comparatively better access to coordinated CF care (Mehta et al. 2010). In *Salty Girls*, thinness and the experience of having CF itself are never infantilized. Rather, participants with very thin bodies all across the gender presentation spectrum are shown as strong, agentic, and resilient. How subjects perform these qualities does vary by gender presentation. For example, the roller derby athlete who projects an air of toughness and aggression offers a more masculinized view of their body. By contrast, the professional model who projects confidence through a serene expression offers a more feminized one.

On the embodiment front, I am probably a fairly typical example of both having CF and being agender. Like many others with a CF diagnosis, I am very thin. I weigh about 80 pounds at just under 5 feet 4 inches tall. Like many others with an agender identity, I do not present as especially masculine or feminine. My style, interests, and behavior span a large spectrum of gendered locations. Sometimes these attributes do indeed lead me to find much common ground with other people with CF diagnoses and/or agender identities, and sometimes they do not.

Here I must reaffirm an important caveat about both CF and agender identity: Not everyone with either or even both of these attributes is thin. Being thin is more common in the CF population than in the general U.S. population because of the particular ways in which the disease often, but not always, impacts the functioning of the stomach, intestines, and pancreas (Stallings et al. 2008). Even in people who do have a high degree of impact on all three of these fronts as I do from my own CF, this impact still exerts itself relative to that person's individual baseline (Sinaasappel et al. 2002). People who have met my mother rarely express surprise at me being extremely thin. My mother, who does not have CF herself or even carry any of the genetic variants commonly associated with it, comes from a long line of skinny people with relatively quick metabolisms. Likewise, I know other people with CF who come from families with generally larger frames. They may be somewhat smaller than their other family members in some cases, but still more solidly built than many of their peers who do not have CF. As treatment options for CF continue to change, some people are also benefiting from drugs that directly modulate the impact of their underlying genetic mutations. This has resulted in some patients finding it easier to gain and maintain body mass (Pettit and Fellner 2014). At present, I myself am not eligible for any of these treatments because I do not have any of the specific genetic mutations that would qualify me. This itself stands as further testament to the diversity of the CF community.

The same diversity exists in the agender community: there are fat people, thin people, and people who do not fall neatly into either of those broad categories (Kozee et al. 2012). However, agender people are often stereotyped as being thin, even though agender identity and slenderness do not necessarily go together. This stereotyping likely stems from general valuation of masculine ideals, and the resulting application of these ideals to things that are intended to be gender neutral (Sykes 2011). Although both agender people and CF patients are frequently stereotyped as being thin, as well as white and youthful, these populations are actually much more varied and complex in their embodiment. Likewise, this diversity highlights the importance of not making assumptions about other people's gender or sex identities. In affirmation of the fact that people's sex and gender identities may not be discernible simply from appearances or casual interactions, I use open terminology (e.g., "female-looking") throughout this manuscript. Such juxtapositions of beliefs about the CF and agender communities with the lived experiences of their members highlights the importance of amplifying diverse voices on embodiment in scholarship on illness and identity.

Consequently, these explorations of how my own lived experiences of CF and agender identity may both converge with and differ from others with these attributes also gives way to a note on methods for this manuscript. Although this paper engages elements of autoethnographic inquiry, it also

presents perspectives on the experiences of others both within and outside the intersection of the CF community and the nonbinary one. Rather than being a methodologically intensive autoethnography itself, this article instead uses autoethnographic inquiries as anchor points for exploration of both prior academic literature and the more public oriented ways in which other people have narrated experiences related to CF and gender nonconformity. I thus refer to it as a "guided review," and commend this approach to other scholars seeking to center multiple standpoints simultaneously in reflection on prior research.

## 2. Literature Review

Because of the many ways CF may manifest visibly on the body, research literature on the disease has often incorporated content on embodiment. Early publications on the intersection between CF and illness embodiment offered a mixed array of benefits and drawbacks for the community. On the one hand, these manuscripts afforded insight into the physical challenges faced by people with CF, and offered context for why patients often look different relative to our peers without the disease (Williams et al. 2007). On the other, these same publications may also have reinforced narrow stereotypes of what people with CF can look like, and even who can have the disease in the first place (Wailoo and Pemberton 2008).

Narratives addressing the relationship between CF and embodiment also appear outside of research publications. Memoirs and essays were among the first venues to make individual accounts of embodiment visible and accessible to people without the disease (Charon 1989). Weblogs soon followed in a similar tradition with the advent of the Internet. Social media groups that are open to the public also give people without CF opportunities to access information about patients' individual experiences of embodiment (Macdonald 2006). Indeed, these patient-initiated narratives have often helped to make research inquiry on CF and embodiment more inclusive and thorough (Petersen 2006).

Similar patterns have appeared with narratives of nonbinary gender experience, challenging research literature to keep pace with a rapidly diversifying pool of personal accounts outside of formal scholarship. As with the academic literature on CF, research on nonbinary identity has undergone a series of evolutions in amplifying the diverse experiences and voices of this community (Richards et al. 2016). This work began with early research on crossdressing (Mason-Schrock 1996) and other gender nonconforming behavior patterns and identities (Serano 2007). Such work has since expanded to explore the variety of ways people across the spectrum of transgender, nonbinary, and gender nonconforming identities make sense of, present, and manage their identities, emotions, interactions with others, and embodiment projects (Sumerau et al. 2016).

Fully amplifying the voices of people living with chronic conditions or any other marginalizing characteristic requires a life course approach. In this regard, the research literature on CF embodiment remains very much in a developmental phase. Interest in experiences of aging with CF (Nowakowski and Sumerau 2019) and gendered embodiment in CF patients (Willis et al. 2001) has become increasingly mainstream in both the scientific and clinical communities. Reflecting this, the 2018 BreatheCon event, coordinated by the CF Foundation, offered sessions focused on aging for the first time ever. The introduction of these sessions followed on the heels of a major milestone in survival being reached the previous year. In 2017, people over 18 with CF outnumbered those under 18 with the disease in the United States. Yet as many key advocates have pointed out, many people with CF still die well before age 30, even if they do not appear especially ill to others. The deaths of prominent advocates Claire Wineland and Storm Johnson within days of each other highlighted this persistent truth (Esiason 2018).

When I facilitated one of the aging sessions myself for BreatheCon 2018, participants' awareness of and conflicted feelings about many of the recent deaths of advocates in their early 20s featured prominently in the dialogue. Our discussions focused strongly on the dissonance between a youthful appearance and the encroaching specter of death. The lack of an intensive focus on aging topics within the general intersection of CF and embodiment thus remains both understandable and problematic.

Because the research literature on experiences of CF past early adulthood remains narrow to begin with, it has thus far failed to incorporate explicit content on the experiences of gender nonbinary individuals living with the disease.

This poses an especially difficult challenge for scholars studying CF and patients' experiences of embodiment because many people often do not achieve a fully actualized gender identity until well into their adult years (Fausto-Sterling 2012). Even as some children gain more options for exploring transgender identities at younger ages of late, the vast majority of people come to an understanding, as well as resources for making sense, of their gender identities as part of early and later adulthood (Meadow 2018). Despite an increasingly developed awareness of how and when people come into nonbinary gender identities as they age, both generally and within the specific context of chronic illness, formal scholarship on nonbinary experience with CF and its relationship to embodiment remains largely nonexistent.

Yet awareness of gender diversity in the CF community is growing. Within the last few years, groups for trans and nonbinary patients have begun appearing on social media, along with more general communities focused on LGBTQIA populations. These groups arose largely in response to connections that people with CF who participated in larger social media communities forged. I vividly recall my first experience of meeting another patient who identified as both nonbinary and queer. Both of us expressed immense excitement and relief at connecting with another person we felt could understand our gender identity as well as the shared experience of having CF.

My inspiration for this manuscript came from observing these general trends in prior literature on CF and then later using them to contextualize two interrelated experiences. First was a sustained history of interactions involving CF, embodiment, and gender. Second was an acute incident that illustrated gaps in our collective understanding of the connections between these three constructs. In critically analyzing that one specific incident, I rapidly progressed to generating copious field notes about experiences from my own personal biography and the lives of my peers in various CF social media and advocacy communities.

## 3. Results

### 3.1. Scripting Thinness through a Gendered Lens

The incident that tipped me into developing a manuscript of my own about CF, gender, and embodiment went approximately as follows. One morning as I was heading to work, I stopped to buy a cup of tea at a convenience store that I used to visit frequently. As I was paying for my beverage, the other clerk—the one who was not ringing up my purchase—fixed me with a stern look and asked "Why are you so skinny?" Both the tone and body language of the cashier, a male-looking person of apparent South Asian descent who appeared to be around my age, conveyed derision. The other cashier, a female-looking person of apparent African descent, looked mortified and told me I did not have to answer. However, I saw an opportunity to educate. Therefore, I looked the other cashier in the eyes and said "That's a shitty question to ask someone." My words were met with silence, so I offered an explanation. "I have cystic fibrosis," I said, fingering the medical ID bracelet around my left wrist. "CF is the most common fatal genetic disease among people with European ancestry. Our median life expectancy is about 47 years right now."

Usually references to potentially terminal illness shut down an awkward conversation with relative speed, or at least prompt a stammering apology and a change of subject (Gramling and Gramling 2012). However, this person chose a different route, and in the process fell back on heavily gendered scripts of how female-looking people experience our bodies. "Well, you know, I didn't know," he scoffed at me. "You women like to work out and everything." I just stared at him, momentarily speechless. Then he added, "But you still look good—I wasn't saying you're ugly or anything." This last was said as if my appearance were my main concern, a classic tactic used to demean and minimize the voices of female-looking people both individually and collectively. Still staring at the cashier, I responded, "No, I just have to take pills

every time I eat because the disease has damaged my pancreas too much for me to digest my own food. The only thing I'm working at is trying not to starve."

The exchange then took a familiar turn that many female-looking readers will recognize quite viscerally: the harsh rebuke waiting on the tongue of a street harasser whose catcalling does not elicit the desired response (Filipovic 2007). "If you want to take it like that," said the cashier, glaring at me. His coworker, at this point, had her head buried in her hands and was whispering: "shut up, shut up." I took a more direct approach, spreading my arms wide and hissing: "If I want to take it like that? Really?" I sucked in the biggest breath I could and then shouted: "Are you fucking kidding me!" before removing my credit card from the other register and leaving without a second look. Some of the other customers in line applauded as the door swung shut, and I could hear the other cashier berating her coworker for being "such an idiot."

This encounter crystallized an acute awareness of the general lack of education about CF and embodiment among those without the disease, gendered expectations of how I will prioritize managing the feelings of others, and the unfortunate intersection of these two phenomena. Even once I had explained that I had a life-threatening illness, the convenience store clerk continued to treat me as if I were violating the entitlement of the male gaze to a less sick-looking body (Nowakowski 2016). When I responded adversely to his behavior, he then acted as if my aggressive response to the situation were strange and inappropriate (Richardson and Green 1999). In the process, he also reduced my own allotted emotional range to a superficial concern for whether or not he found me attractive (Marcus 2013).

This illustrated both the generalized dynamics of deriding the bodies of the many people with CF whose bodies show common consequences of our illness (Nowakowski 2016), and the gendered expectations people often hold about how we should respond to such comments (Richardson 2005). My male-looking peers with CF have often told stories of interactions that began similarly, but had different outcomes when they responded with a similar degree of indignation. The incident at the convenience store affirmed evidence from the literature that female-looking people with chronic conditions are often expected to respond to derisive comments about our bodies by seeking affirmation that we "still look good" (Bartky 2002), whereas male-looking people are expected to respond with anger when insulted (Hess et al. 2005).

As an agender person, I did not spare a thought for whether or not my own behavior in this scenario was "unladylike" or "overbearing" (Hess et al. 2009). Rather, I focused on the discomfort of the other person attempting to force me into gendered scripts for my thinking that have never matched how I actually experience my own body (Young 2005). This specific incident thus provided a flashpoint for organizing a much broader array of observations and insights about the relationships between CF, gender, and embodiment. It did so by sharply highlighting the often contradictory ways in which people react to my body, both through the individual lenses of health and gender and through the intersection of the two frames. In the process, it illuminated a variety of ways in which my own agency becomes subordinate to rigid norms of how I should experience my own body and the illness that envelops it, and provided context for the liberation I find in spaces that simultaneously center my identities as a CF patient and agender person.

### 3.2. On Food and Eating

A common manifestation of contemporary American society's pervasive assault on the agency of female-looking people is the recapturing of perceived control through rigid behaviors surrounding food and eating (Ata et al. 2007). "What do women eat?" is often a question with no easy answers because women face a lot of competing pressures surrounding food and eating (Strong et al. 2000). These pressures vary by culture and context within the United States, but have several common features (Banner 1983). Adults who look female are expected to be slender, although not so thin as to appear overly childlike or otherwise nonsexual in the eyes of others (Brumberg 1998). Likewise, we are expected to appear fit and active without being overly muscular or burly (Bordo 2004). Thinness also signals control of oneself and one's desires, and thus captures elements of the overall expectations for

women to behave subordinately (Brown and Jasper 1993). This introduces an added level of social complexity for female-looking people with CF who do not identify as women, or indeed with any binary gender. We experience a lot of the same pressures that cisgender women do surrounding food, weight, and related topics (Bowman 2018). On top of that, we experience cognitive dissonance about the gendered ways in which these concepts are applied to our own lives by outsiders (Warwood 2016).

Being a thinner person even by standards adjusted for my CF diagnosis, I often experience a variety of frustrating and contradictory feedback concerning my eating behaviors and their perceived connection with my physical appearance. People constantly point out, as in the example above, that I am very thin. Sometimes the language is merely pitying: "frail" or "waiflike." Sometimes it takes a more cutting tone: "skeletal" or "emaciated." I have been told by many people over the years that I look "fragile." The degree of explicit gendering present in these reactions varies, but there is often an underlying implicit message that women should look a certain way (Murnen and Don 2012). This also raises questions about the problems inherent in assuming that someone identifies as a woman based on appearances alone.

The double jeopardy faced by both those who actually are women and those who are merely perceived as such by others becomes even clearer in the ensuing portions of such exchanges. If I am not in a situation where I am actively eating something, this infantilizing and gendering response to a thinner female-looking body generally segues into people telling me I need to eat more. In some cases, this involves trying to push food on me. I have written in the past about people grabbing my grocery cart at stores and trying to put additional items into it (Nowakowski 2016), a perplexing behavior given that I actively choose foods rich in caloric energy and monounsaturated fats. Furthermore, if I *am* in a situation where I am actively eating, I get mocking comments for eating more than a tiny amount at a time. No matter how much or little I might put on my plate during a given meal in a public space, someone will invariably comment on it. When I diligently follow national guideline recommendations for my energy intake, people—and almost uniformly male-looking ones—mock me by suggesting I cannot possibly eat "all that." When I take only a small amount, the same people tend to assume that I have an active eating disorder, and rush to reassure me that I "still look good" (Charmaz and Rosenfeld 2006).

Leaving aside the massive flaws in this broader public understanding of disordered eating, a phenomenon with which my own past experience can hardly be separated from my CF diagnosis and its gross mismanagement in my younger years, these occurrences show a clear pattern of constraining the agency of female-looking people in feeding ourselves (De Groot and Rodin 1994). As a person with an agender identity, I find some comfort in my own internal freedom from concerns about whether or not people perceive my eating habits as somehow dissonant with femininity (Strong et al. 2000). However, my female-looking body renders this comfort a neutral space at best because it still prevents me from accessing any actual social rewards for following a guideline-based diet (Leavy et al. 2009). By contrast, my male-looking peers with CF often get supportive comments and other positive reinforcement for eating large amounts of food at once (Ruby and Heine 2011). Following the recommended high-calorie, high-fat diet for CF patients (see (Stallings et al. 2008)) becomes one of many ways that male-looking people with CF, even ones who are very thin, can perform and affirm masculine identities. This practice is not unique to people with CF, but rather a broader way of signifying legitimate masculinity in the contemporary United States (Darwin 2017a). Being able to eat a large amount of energy-dense food gives male-looking people access to valuable social capital and thus privilege (Darwin 2018), whereas it more often exposes female-looking people to judgment and stigma.

My own understanding of these phenomena has historically been reinforced by observing that adverse reactions by others to my own food behaviors occur markedly less on days when I dress more androgynously and/or have my hair cut shorter. On days where I present myself physically in ways that center the dualism in my own gender identity more, I experience fewer confused reactions from others to food and eating behaviors that are specific to my CF. Yet days like these come at a price in the form of

confused and even threatening reactions to my gender presentation itself. Because my body is viewed as female-looking rather than male-looking, these threats usually do not come to any sort of fruition (Burdge 2007). For agender people with CF who look male but present with more femininely coded styles, a different set of concerns may arise. These additional concerns may likewise be partially mitigated by having a thinner body, which itself invites interpretation as being more feminine (Pine 2001).

### 3.3. Dualistic Approaches to Aging

Such contradictory responses to how gender nonbinary people with CF experience and respond to our bodies extend far beyond food and eating (see (Lucal 1999)). Interactions related to aging may represent another common node around which dualistic experiences related to embodiment tend to cluster. I refer to this phenomenon in a more general sense as "the 'young lady' paradox," which may well become the focus for its own manuscript in the future. In the context of this broader synthesis of scholarship and experience on CF, gender, and embodiment, this paradox involves a layered series of very particular ironies.

First of all, I am not that young to begin with. At 34, I am aging out of the "young adult" stage of life, although not yet perceived as "middle aged" based on chronology alone (Elder et al. 2003). Within the context of my CF diagnosis relative to my birth cohort, I am actually somewhat advanced in age. The average life expectancy when I was born was much lower, and remains substantially different from statistics for the general U.S. population even now (Cohen-Cymberknoh et al. 2011). This disparity is even worse for patients assigned female at birth (Davis 1999). When I was tentatively diagnosed with the disease at only a few years old, CF was still largely considered a childhood disease (Elborn et al. 1991). At the time of my conclusive diagnosis later on, my chronological age was a mere five years below the average life expectancy for U.S. residents with CF.

Yet if people perceive my age as being different from the actual quantity of years I have lived, they tend to do so in a downward direction. This likely relates back to content from the prior section on food and eating related to the viewing of thinner bodies as being somehow lacking in agency or strength (Holubcikova et al. 2015). Within this context, the physical evidence of my CF ages me down in the consciousness of others. This phenomenon may be exacerbated in my specific case by general patterns of perceiving people with a more neutral or mixed gender expression as being more youthful (Fraser 2017). Rigidly gendered behavior gets encouraged quite widely in childhood, but violating these expectations does not tend to meet with the same punishment during earlier portions of the life course (Galambos et al. 1990). These dynamics then intertwine in a simultaneously causal and consequential way with the inaccurate stereotype that only younger people hold nonbinary gender identities in the first place (Frohard-Dourlent et al. 2017).

As a female-looking agender person with CF, I also experience some particular nuances of these more general phenomena that speak to broader social perceptions about gender and aging. When people express surprise at my age, they generally indicate a belief that they are complimenting me (Lauzen and Dozier 2002). This almost certainly stems from a widespread belief in the U.S. and other countries of similar culture that female-looking people and those who identify as women fear aging as both an overall process and a series of specific physical changes (Monaghan 2001). Literature across a variety of disciplines describes linkages between the aging process and fears of being perceived as unattractive, powerless, nonsexual, weak, and dependent (Calasanti and Slevin 2001).

Indeed, suggesting that I should value being perceived as youthful represents an accurate framing of the pressures that female-looking people often face (Calasanti and Slevin 2013). My agender identity renders this framing somewhat problematic in a broader social sense. Much greater dissonance comes from the simple fact that for people with CF, getting old is a privilege historically afforded to very few (Kulich et al. 2003). Current survival data on CF suggest that those people with the disease who *do* live to very advanced ages are substantially more likely to have been assigned male at birth (Milla et al. 2005). This represents yet another gendered way in which people with CF may experience

our bodies differently as we age, and how those nuances may collide for those of us with nonbinary identities (Darwin 2017a).

### 3.4. The Female and the Function

Copious literature in the interdisciplinary sociomedical sciences already explores the expectations related to embodiment that female-looking and woman-identified people face (Moss and Dyck 2003). This research identifies several intersecting issues that contribute to the oppression of female-looking patients in both clinical and community settings. These include, but are not limited to: essentialist ideas about the relationship between a body's appearance and its purpose, conflation of sex with gender, binary notions of both sex and gender, disregard for reproductive autonomy, and devaluing of feminine qualities (Puopolo 2018). Female-looking people with nonbinary gender identities still experience the harms of these phenomena, with additional nuances introduced by the dissonance others perceive between our physical appearances and our cognitive selves (Darwin 2017a).

It has thus never surprised me to reflect on the often divergent experiences of my male-looking peers within the CF community. Even those male patients I know with nonbinary gender identities often have very different histories with gendered expectations than I myself do (for examples of nonbinary diversity beyond CF communities (see again (Darwin 2017a; Mason-Schrock 1996; Serano 2007; Sumerau et al. 2016)). The aforementioned example of how male-looking people with CF may experience very different social responses to following a guideline-based diet constitutes only one illustration of these differences.

Even staying within the realm of food and nutrition, a variety of gendered nuances in the experience of CF embodiment present themselves. One key advocate I work with, who identifies as both male and a man, spends most of his leisure time on athletic pursuits. He was raised by a professional athlete and came to value sports in his own life both for the joy they bring him and for the ways in which they facilitate keeping his body healthy. Because the ability to stay active in sports was extremely important to this person, he elected many years ago to get a feeding tube implanted. He now speaks very openly in the public domain about how much he values the tube as a resource for helping him achieve his goals in sport (Esiason 2015). The language he uses to discuss his tube and its impact on his life reflect a sense of identity around athletic achievement, a theme that commonly appears among male-looking and man-identified people in the U.S. (Coakley 2006). Indeed, people who look male are often pressured to pursue athletics even if they do not identify as men, and sanctioned if they do not comply (Wiley et al. 2000).

As suggested by the example above, these differences are not always uniformly positive. In the case of my one colleague who does identify as a man, gendered expectations surrounding athletic activity have likely exerted some positive influences by making him want to challenge himself in sports and reap the health benefits of doing so. However, both male-looking CF patients who identify as men and those with other gender identities may experience harm from the idea that one of their key purposes in life is to excel in sports. The idea that bodies with a certain appearance should perform certain functions can damage as easily as it can inspire (Fallon and Jome 2007). A good friend of mine in the CF community who identifies as a genderqueer male often tells me that he feels insecure about his body looking weak because it is small and thin. This is a common concern among male-identified people (Leit et al. 2001). Being assigned female at birth, I probably spend less time worrying about these kinds of concerns.

Conversely, I have faced a slew of other challenges to my self-concept and ability to articulate it to others that my male-assigned peers often have not. Just as athletic pursuits are considered a marker of virility and thus worth in male-looking people, similar pressures apply to female-looking patients with respect to physiognomy (Tylka and Calogero 2011). Whereas U.S. residents who are assigned male at birth are often expected to become sturdy and muscular as they grow, those assigned female face similarly strong expectations to stay slender, but not too slender (Paechter 2006). Feminized expectations about embodiment place constraints on people at both ends of the size

spectrum, often leading to perceptions of very thin female-looking patients as desexualized and unfit for reproduction (Lupton 2012). The latter has never bothered me as an individual, both because I do not see myself as a woman and because I have never taken interest in having children by any means. The former, however, has impacted me both independently and in intersection with negative perceptions of my more ambiguous gender presentation. I have heard all of the following comments about my body at moments that should have been very intimate and fulfilling: "you're disgusting," "this is like having sex with a skeleton," "you kiss like a man," "you're too dominant," "you're too sexual for a woman," "I don't want to talk about your disease," "you're lucky I want to be with someone who looks like you," "it's not sexy that I can see the sinews in your stomach," and "you never want to have sex." I refer broadly to this history as "Schroedinger's womanhood"—simultaneously not being a woman intrinsically and punished for not being enough of one extrinsically (Nowakowski 2016). This experience is not uncommon among CF patients as a whole (Willis et al. 2001), but may have unique nuances for those with nonbinary identities.

Perhaps unsurprisingly, I have found liberation and comfort in building relationships with other CF patients who simultaneously identify as nonbinary, kinky, and polyamorous (see (Willis et al. 2001)). I have found tremendous understanding within these spaces, which include some of the aforementioned social media groups centering non-cis and non-hetero experiences. I have found resonance in these spaces surrounding both my generalized lack of concern about my body matching a gender identity I do not have and my specific perceptions of how I am constrained by others attempting to define gendered roles for me (Julia 2003). I have also found community and liberation with people in queer- and trans-centering CF discussion groups who incorporate kink practices into their lives (Banerjee et al. 2018). These practices are often extremely diverse across patients; the "kink" moniker can describe a wide variety of activities and interests (Newmahr 2010).

Masochism, often one of the first things people think of when hearing the word "kink," certainly features in the lives of many people with CF who negotiate embodiment through these channels (Williams et al. 2016). Documentarian Kirby Dick explored specific connections between CF and kink in *The Life and Death of Bob Flanagan, Supermasochist.* The performance artist profiled in the film had a male identity, but a "deviant" one that afforded him freedom from excessively gendered expectations of what he would be and do (Banerjee et al. 2018). By practicing masochism in public forums for consumption as art, Bob Flanagan both articulated an authentic gender identity that made him feel more centered in his body and achieved a sense of being able to escape the physical challenges of living in that body (Reynolds 2007). He used kink in a life course context to preserve mastery of a non-mainstream masculinity as his body changed from and eventually succumbed to the progressive damages of CF (Reynolds 2007).

*3.5. Transplant and the Gendered Body*

The subject of organ transplantation, a common theme in the biographies of adults with CF in the U.S., also appeared prominently in Bob Flanagan's work (Reynolds 2007). Indeed, transplantation constitutes a major content node in both Flanagan's own performance art (Reynolds 2007) and that of other patients following in his footsteps (Lowton 2003). His work has inspired a wide array of subsequent performances engaging kink by other artists with CF (Reynolds 2007), including a good friend of my own who recently underwent a double lung transplant. These experiences illuminate both the overall process of adjusting to a changed body (Lowton 2003) and the specific processes of renegotiating that body in relation to one's gender identity (Riessman 2003). In Flanagan's case, transplantation required him to revisit his sense of masculinity and consciously work to continue cultivating his identity as not merely a masochist, but a supermasochist, one who could best any other (Reynolds 2007). This appears as a common trope in notions of what makes a man successful both overall and within trajectories of chronic illness (Riessman 2003).

In my friend's case, transplantation sparked a different journey surrounding gender and embodiment. His work as a genderqueer performance artist with CF and a double lung transplant

explores themes of gendered embodiment through the lens of a masculinity that is not only kinky, but also nonbinary (Riessman 2003). He explicitly cites Bob Flanagan's work as an inspiration for his own artistic ventures (Reynolds 2007), and also points out how his work differs from these early public demonstrations of negotiating illness embodiment through kink practice. He has also described in our discussions how Flanagan's work pushed him to exercise creativity in imagining the post-transplant body in physical space and as a dynamic entity (Lowton 2003) as well as a gendered one (Willis et al. 2001). In his own work, he focuses less on the obvious choice of highlighting the scars introduced by the transplant surgery, and more on the rest of his body and how it behaves.

Even the territory of scars themselves affords ample opportunity for both negotiating embodiment and highlighting the gender diversity of the CF community within that context (Lowton 2003). Indeed, *Salty Girls* centers a variety of themes related to transplant scarring, and its relationship to gender identity and presentation (see (Willis et al. 2001)). Some of the narratives for the project talk about adjusting to scars and the idea of a marked-up body as unladylike (Pettigrew 2012). Some patients love this because their gender presentation more closely matches their identity after transplant, whereas others hate it and feel alienated (Pettigrew 2012).

Both discussions with friends in the CF community and explorations of broader representation projects have helped me to reflect on my own experiences with surgery and scarring. As of yet, I have not undergone organ transplantation surgery (Nowakowski 2018). Although it appeared likely in my early 20s that I would eventually need a lung and/or kidney transplant, changes in my treatment and my positive response to those alterations appear at present to have bought me substantial time (Nowakowski 2018). I remain both open to transplantation in the future and conscious of how it would require me to renegotiate my body in both general and gendered context (Nowakowski 2018). That said, complications from my own CF have already led me to undergo a variety of other surgeries that acclimated me to the process of adjusting to dramatic changes in the appearance of a particular body part (Nowakowski and Sumerau 2019) and the unique dynamics of those negotiations for me as an agender person (see (Willis et al. 2001)).

The multiple surgeries I had to rebuild my gums, which along with my teeth, had been almost completely destroyed by nearly three decades of chronic infections, proved especially instructive in this regard. I began this journey with teeth that were either mostly or entirely synthetic above what remained of my gumline, and exposed bone at the base of the remaining natural roots. A dental provider once described my mouth as "unsightly" and asked: "you want to look better, don't you sweetheart?" Sitting in that office, I wondered if my dentist would have asked the same question of a male-looking patient (Moss and Dyck 2003). I shrugged and said that I cared more about the functioning of my mouth than its appearance or how that might change over time (Barrett 2005). The conversation then shifted to the substantial risk that I would lose what remained of my natural tooth material, and likely multiple sections of my jawbones, to necrosis if I did not undergo surgery to rebuild the gums and protect the underlying structures. This got my attention, and also made me wonder why the dentist had not led with this information. Lacking any concern for whether the inside of my mouth looked unfeminine or might be perceived as such by others, I found it appalling that a health provider would think that I might care more about aesthetic concerns than about losing portions of my face to infection (see (Moss and Dyck 2003)).

Because I very much did not want to weather the agony of my face literally eating itself, I scheduled a surgical consult. This gave way to a series of five surgeries over an 18-month period, involving the following interventions: removing all four of my diseased wisdom teeth, harvesting connective tissue from the roof of my mouth, stitching the harvested tissue down around the exposed roots of my remaining teeth, stretching the grafted tissue to extend it, and repeating the process (Nowakowski and Sumerau 2019). Each recovery involved several days of spitting out surgical plastic and rubber sutures caked with blood and thickened mucus. The grafted tissue remained white as the bones beneath until it slowly grew blood vessels and transformed into something resembling a "normal" gum. My own sense of normalcy had long since escaped me by that point, and I watched the

gore inside of my mouth as one might a particularly compelling horror film. However, I never found it ugly, and took a bizarre delight in relaying all this to my students each time I would return to campus after a procedure. Their stunned reactions made me feel more authentic in my body as an agender person, rather than concerned for potential harms to my womanhood or insufficient bolstering of my manhood (Darwin 2017b).

*3.6. Performing Toughness*

My experience with gum reconstruction surgeries thereby illustrates the role of toughness in the embodiment of CF (Willis et al. 2001). I use my visible scars from the operations as a means of illustrating both the damage the disease can do and my own resiliency in coping with those harms (Lowton 2003). When I peel my lips away from my teeth to show what I refer to as my "Frankengummies," I invite people to engage with my body in a way that involves both trust and challenge. I communicate that I feel confident that they will not mock me, and also that I am strong enough to withstand anything that they might have to say in response. In these moments, I feel keenly aware of the unique dynamics of extending trust to others while occupying a female-looking body, and also of my total disregard for whether people see me as more or less feminine for having stitched-together gums that still resemble something one might see in a cadaver lab.

As an overall construct, toughness is centered in a broad array of narratives about CF across both gender and sex groups. For example, people who followed the activist Claire Wineland prior to her death often cited her bravery and brashness about facing mortality at an early age (Denaro 2016). Her YouTube videos featured activities like escaping a hospital to attend a political rally at the beach, with a sly but thoroughly earnest emphasis on her own desire to be "hardcore" about getting the most out of life with CF. Yet Wineland's videos also showed her applying a distinctly and deliberately feminine touch to the spaces she occupied: soft paint colors and fuzzy throw pillows in her apartment, and string lights and photo collages in hospital rooms. Her own performances of toughness communicated a desire to be seen as simultaneously strong and womanly as she came of age into adulthood (Denaro 2016). I have met other people with CF who perform toughness through competitive bodybuilding and extreme sports (see Willis et al. 2001), and of course the aforementioned "supermasochist" practices (Reynolds 2007). Indeed, the question of the body as narrative art pervades every approach that patients use to negotiate toughness in our own lives (Lowton 2003).

Therefore, perhaps the most important lesson from this exploration is a broader one about the relationship between the gendering of bodies and the narration of them. As an agender person with CF, I find that I experience my body in gendered ways when others attempt to control the narrative of why it appears the way it does (Gergen 2000). This happens in different but equally destabilizing ways with people who are unaware that I have CF and with people who know about my diagnosis. Much of the literature on progressive disease suggests that mentioning a diagnosis can kill a conversation, especially one of an adverse nature, thereby allowing the person dealing with health challenges to feel empowered and agentic once more (Gramling and Gramling 2012). Yet my own experiences suggest that other people's interest in gendering my body sometimes outstrips their sense of social appropriateness surrounding my life-threatening chronic disease (Willis et al. 2001).

Future study of chronic diseases and gendered bodies should thus explore more nuanced dynamics in who controls the narrative of embodiment, and under what conditions this occurs. I recently resonated with an account of health embodiment nested within a broader exploration of career development in hip hop music: Jooyoung Lee's *Blowin' Up*. One of the rappers Lee follows in his ethnography, Flawliss, recounts his experiences of recovering from surgeries and returning to musical performance after getting shot. Lee relays how Flawliss initially used anecdotes about the shooting and his recovery to build himself up before battle performances, and to impress fans in conversation after leaving the stage. However, Flawliss felt his control of the narrative of his own body shift when a fan asked a very blunt question about his colostomy bag (Lee 2016). In the same way that the shopkeeper's questions about why I am so skinny shifted my own control of my body's narrative,

so too did the fan's inquiry make Flawliss feel disempowered in crafting his own story. He recalled feeling weakened and emasculated by the exchange (Lee 2016).

I found the juxtaposition of Flawliss's message of strength, bravery, and resiliency with the fan's perception of him as weak, damaged, and compromised both powerful and relatable. I tell many similar stories in my teaching and advocacy activities (Nowakowski and Sumerau 2015). I might recall my many surgeries done with only partial sedation, and the experience of watching a bloody scalpel emerge from my own mouth; or I might mention casually how it feels to cough up a mucus plug during a work meeting, and have to excuse myself to the bathroom to spit the bloodied mass into a paper towel. As an agender person, I am aware of how readers might interpret the first story as a performance of masculinity and the second as one of femininity beneath their overall character as social constructions of toughness (Evans et al. 2011). I am scarcely alone among CF patients in performing this kind of grit, whether in deliberately gendered ways or otherwise (Lowton 2003). *Salty Girls* alone provides ample examples of how people living with CF turn our medical devices and the visible ravages of our condition into sign equipment that communicates our strength. Indeed, some participants wear their oxygen tubes and tanks almost like armor, posing defiantly for the camera (Pettigrew 2012). In some cases, this posturing coheres with the subject's overall apparent gender presentation; whereas in other cases, it diverges.

The performance of toughness surrounding visible evidence of CF in *Salty Girls* also prompts the question of what happens outside of the artistic environment, when people on the street make invasive comments or act as if patients cannot do anything for ourselves (Kane et al. 2005). Moreover, it demands reflection on how broader gender scripts intersect with these behaviors; for example, the idea that "chivalry" dictates not allowing female-looking people to carry things for themselves (Ridge 2014). For nonbinary people with CF and other disabling conditions, a forced gendering of our bodies in completing everyday activities may bring extra layers of disempowerment (Willis et al. 2001). When a person grabs my bags or even openly expresses incredulity that "such a frail little thing" could possibly function independently, I experience my body in ways that are both inappropriately gendered and distinctly negative (Nowakowski 2016).

Repudiating such problematic narratives often results in further gendering, as seen in the example from the convenience store. An occasion on which I told an interloper that I could carry my own bags comes to mind. This interaction began with the person intruding into my personal space, grabbing my grocery cart, and beginning to rifle through my bags despite me telling him to back off immediately. What began with "let a gentleman help you" quickly devolved into a demonstration that this individual was anything but. The interaction ended with the other person scornfully asking "What if I were a woman, huh?" as if to imply that I were being sexist by pointing out the inappropriateness of grabbing a female-looking person's bags without their consent. In so doing, the other party uncannily mirrored evidence from the literature about the symbolism of bags in gendered interactions (Ridge 2014). The lack of self-reflection inherent referring to another human being as a *thing* aside (Nowakowski 2016), the other person's responses to me used gendering as a form of dehumanizing and decentering someone who had made him feel inadequate as a man (Ridge 2014).

## 4. Discussion

Gender and its relationship to embodiment are deeply contextual, both within and outside of the CF community. How individual CF patients experience these constructs are thus unique, whether or not they identify as nonbinary. Prior literature on gender, embodiment, and chronic illness offers context for why not every nonbinary person with CF experiences their identity the same way. Even if they share the specific experience of being agender, people's illness biographies and resulting relationships with their own bodies may differ dramatically (Charmaz 2000).

My experiences as an agender patient also frequently diverge from those of people with CF who are genderfluid or have a more static liminal identity such as androgynous. This may owe in part to the fact that gender identities lying in between the two poles of an established binary, rather than

wholly outside of it, appear to correlate with more frequent and intense experiences of dysphoria (Darwin 2017a). Because people with CF often experience conflict in our relationships with our bodies to begin with, having a more liminal nonbinary gender identity may introduce additional complexities into the negotiation of embodied illness identity (Puopolo 2018).

Further nuances almost certainly factor into my own experiences as an agender CF patient who looks female and identifies as such. Although my sex identity has always been transparent, it may also play a more latent role in my negotiation of embodied identity as a gender nonbinary person with CF. Specifically, there may be a hidden curriculum of sorts in gendered representations and reactions to CF because a high proportion of highly visible people with the disease are female-looking. This may be true for many reasons, but almost certainly involves more generalized survival dynamics in the chronic illness population that are sociologically conditioned (Bird and Rieker 2008).

Future research on CF, gender, and embodiment should thus focus both on intensive exploration of the experiences of gender-diverse groups of patients with female sex identities and assignments, and on amplification of the voices of patients with other sex identities and assignments. Centering an array of nonbinary biographies in growing the literature on CF and embodiment can offer unique perspective on how people living with the disease find empowerment and liberation while occupying bodies that often feel deeply constraining. Indeed, the notion of constraint evokes important themes from prior sociomedical literature on gender and health (Bird and Rieker 2008). Understanding how nonbinary people with CF experience our bodies both within ourselves and through interactions with others affords an opportunity to support intersectionally marginalized patients in achieving effective illness management (see (Charmaz 2000)). Specifically, amplifying these narratives in research and practice can help nonbinary people with CF to achieve a sense of coherence in our relationships with our bodies and how they reflect our identities beyond the disease.

**Funding:** This research received no external funding.

**Conflicts of Interest:** The author declares no conflict of interest.

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
