# Peer review of "The Salt without the Girl: Negotiating Embodied Identity as an Agender Person with Cystic Fibrosis"

_socsci, doi:10.3390/socsci8030078_

Round 1

Reviewer 1 Report

The author presents a personal account of experiences with being a person with CF, agender, and negotiating embodiment across these two identifies. The author does a nice job of using autoethnography to accomplish the goals of the paper and presents a clear and comprehensive view of the issues.

Overall 

There is some extraneous detail or language that can likely be removed while still communicating the ideas (for instance, line 113 about the author's mother's favorite activity being blood donation doesn't contribute to the article). Also, at times as a reader I felt the author was coming across somewhat hostile, especially in the description of the author's own experiences. Although I understand this is about the author's experience, perhaps there may be places where the language can be toned down so it isn't as jarring. 

1. Introduction. 

At times the language can be ambiguous in the introduction section making the reader perceive the author to be less confident in the material. As you read on, you see that is not an issue. I suggest removing terms such as "basically" on line 29 or "somewhat" on line 31. 

Sentence beginning with "In this regard, it is part...(Richards et al. 2016) is confusing to read. 

Although I appreciate the author's attempt to simplify the pathophysiology of CF, calling it a "relatively simple problem" seems like an understatement given that it starts at the DNA level and actually has quite different "problems" across different mutations with varying degrees of complexity.

I think two key pieces of information are missing from the description of CF in lines 47-54. I think it is relevant to mention it is genetic and also its impact on mortality. The author gets to this later on, but an uninformed reader should be made aware of this from the outset.

Lines 64-65. "One of the most visible representation of people with CF..." It is unclear what that most visible representation is. Is it looking female? Or looking thin?

The description of The Salty Girls project on lines 70-71: Is this Pettigrew's goal? Did Pettigrew intend to explore gendering? Or is this the author's interpretation of the work/bi-product of Pettigrew's work?

Reference to higher economic class on line 85. How is higher economic class conveyed through photos?

Lines 94-96: The description of the roller derby athlete and the professional model. It isn't clear to me which one is which in terms of gender presentation, which seems relevant given that I believe the author is trying to make the point that how one displays qualities of strength and resilience varies by gender presentation.

The reader could benefit from an understanding of why agender people are stereotyped as thin. This is unclear, whereas the discussion of why people with CF are often thin is comprehensive.

The discussion of weight, thinness and embodiment in lines 103-116 could benefit from the acknowledgement of how this may be impacted by the introduction of CFTR modulator therapy, which is correcting some of the problems associated with malnutrition in CF. There are anecdotal reports of people with CF gaining weight, even at unsafe rates/level. I imagine this would produce a huge shift in embodiment. 

2. Literature Review

Line 159: Typo with semicolon in parenthetical reference with Serano 2007.

Line 168: use of the phrase "first time ever" makes it sound like it was a neglected area for a long time, but the event was in its infancy when it did introduce this topic. 

3. Results

Line 232. Found this part of the story of the incident hard to follow. Wasn't clear if the cashier actually said "you bitch" or if the author is drawing an analogy. 

Lines 274-276: The sentence starting with "What do women eat..." was confusing. I think there is an opportunity for a description here of what the pressures surrounding food, weight and related topics are.

Line 288: The sentence "To date, no one has bothered...." Would this be an equally offensive question because it implies binary gender. Isn't the larger problem that we shouldn't expect anyone to look a certian way?

Lines 393-404: Is this information about this advocate taken from public domain content published by the person or from personal communication. If the latter, there may be an ethical issue since this person may be easily identifiable. 

Line 413: The author describes a person communication and references a peer-reviewed journal article. Confusing. 

Author Response

Reviewer #1

The author presents a personal account of experiences with being a person with CF, agender, and negotiating embodiment across these two identifies. The author does a nice job of using autoethnography to accomplish the goals of the paper and presents a clear and comprehensive view of the issues.

Thank you for both your positive feedback and your detailed constructive suggestions! I like the new draft of the manuscript a lot better after incoporating your input.

There is some extraneous detail or language that can likely be removed while still communicating the ideas (for instance, line 113 about the author’s mother’s favorite activity being blood donation doesn’t contribute to the article). Also, at times as a reader I felt the author was coming across somewhat hostile, especially in the description of the author’s own experiences. Although I understand this is about the author’s experience, perhaps there may be places where the language can be toned down so it isn’t as jarring.

Thank you! I agree about both the excess details and the language—if anything in the first draft read as hostile, preserving it in the revised version would work against my aims of inviting people into my experiences and encouraging them to explore on their own. Hopefully the revisions I’ve made in response to this and your other specific edits have smoothed out the tone, but I’m happy to make further edits if desired.

At times the language can be ambiguous in the introduction section making the reader perceive the author to be less confident in the material. As you read on, you see that is not an issue. I suggest removing terms such as “basically” on line 29 or “somewhat” on line 31.

Makes sense. I’ve simplified the sentences in early portions of the introduction to read more directly and declaratively.

Sentence beginning with “In this regard, it is part... (Richards et al. 2016) is confusing to read.

Agreed. I’ve rewritten this sentence accordingly.

Although I appreciate the author’s attempt to simplify the pathophysiology of CF, calling it a “relatively simple problem” seems like an understatement given that it starts at the DNA level and actually has quite different “problems” across different mutations with varying degrees of complexity.

I totally agree with you! I explained it this way in the first draft because this is a social sciences journal and I’ve been critiqued by reviewers in those disciplines before for getting into too much technical detail about the disease. I work at a medical school, so suffice it to say I’m more than happy to provide some enrichment about the nature of the disease. I’ve done so in the revised draft.

I think two key pieces of information are missing from the description of CF in lines 47-54. I think it is relevant to mention it is genetic and also its impact on mortality. The author gets to this later on, but an uninformed reader should be made aware of this from the outset.

So true. Peer review always helps me spot what I’ve missed from becoming so accustomed to my own experiences in doing autoethnography. I’ve added these two elements up front in addressing what CF is and how it impacts patients’ lives.

Lines 64-65. “One of the most visible representation of people with CF...” It is unclear what that most visible representation is. Is it looking female? Or looking thin?

I totally rewrote this sentence, as I found it confusing as well on the reread.

The description of The Salty Girls project on lines 70-71: Is this Pettigrew’s goal? Did Pettigrew intend to explore gendering? Or is this the author’s interpretation of the work/bi-product of Pettigrew’s work?

Good questions! I would say the answer is a bit of both. Pettigrew does center diverse representations of gender in all of his photography—not just his work on CF—and explicitly discusses this focus in some of his blog writings and social media posts. I also that I tend to focus centrally on these aspects of his work, and feel drawn to them, in part because of my own experiences. I’ve added these notes to the revised manuscript in my discussion of Pettigrew’s work.

Reference to higher economic class on line 85. How is higher economic class conveyed through photos?

Great question! I’ve added some notes and references from interactionist/dramaturgical sociology to clarify this concept. In a nutshell, economic class is usually conveyed through sign equipment (clothing and other “stuff” that the person has with them) and body language. I elaborate on this a bit in the revised mansucript.

Lines 94-96: The description of the roller derby athlete and the professional model. It isn’t clear to me which one is which in terms of gender presentation, which seems relevant given that I believe the author is trying to make the point that how one displays qualities of strength and resilience varies by gender presentation.

Agree completely! I’ve added some details to clarify these points.

The reader could benefit from an understanding of why agender people are stereotyped as thin. This is unclear, whereas the discussion of why people with CF are often thin is comprehensive.

Strongly agree. I’ve gone back to this component of the literature review and added additional details and citations about social and cultural representations of agender people in the US, and their connection to embodiment.

The discussion of weight, thinness and embodiment in lines 103-116 could benefit from the acknowledgement of how this may be impacted by the introduction of CFTR modulator therapy, which is correcting some of the problems associated with malnutrition in CF. There are anecdotal reports of people with CF gaining weight, even at unsafe rates/level. I imagine this would produce a huge shift in embodiment.

Goodness, yes. And thank you for pointing this out. One thing I’ve clarified as a complement to the added content on modulators is that I’m personally not eligible for modulators because I have very rare CFTR genetics. This surely shapes my sense of how relevant the development of these drugs is for my own experiences and efforts to communicate them to others.

Line 159: Typo with semicolon in parenthetical reference with Serano 2007.

Well spotted. Thank you for doing such a close review! I’ve fixed this in the revised version.

Line 168: use of the phrase “first time ever” makes it sound like it was a neglected area for a long time, but the event was in its infancy when it did introduce this topic.

Good point. I’ve scrapped the “first time ever” language entirely during revisions.

Line 232. Found this part of the story of the incident hard to follow. Wasn’t clear if the cashier actually said “you bitch” or if the author is drawing an analogy.

This was intended fully as an analogy, and has been clarified as such in the revised version.

Lines 274-276: The sentence starting with “What do women eat...” was confusing. I think there is an opportunity for a description here of what the pressures surrounding food, weight and related topics are.

Agreed. I’ve provided this content as an extension of my response to Reviewer #2. Specifically, I’ve incorporated this suggestion as part of a generally deeper engagement of feminist literature on embodiment and eating.

Line 288: The sentence “To date, no one has bothered....” Would this be an equally offensive question because it implies binary gender. Isn’t the larger problem that we shouldn’t expect anyone to look a certain way?

I would give a resounding “yes” to your question here. I think the way I phrased this sentence in the original manuscript communicated a message different from what I’d envisioned in writing it. So I reworded it completely as a reflection on the fallacy of making assumptions about people’s gender based on appearances.

Lines 393-404: Is this information about this advocate taken from public domain content published by the person or from personal communication. If the latter, there may be an ethical issue since this person may be easily identifiable.

Indeed it is, and that’s why I provide citations for this advocate’s blog in incorporating the content. I love that you raised this question!

Line 413: The author describes a person communication and references a peer-reviewed journal article. Confusing.

Clumsy writing on my part! I’ve revised this sentence into two distinct ones, the first describing the communication and the second noting it as an example of the phenomenon explored in the article I cited here.

Reviewer 2 Report

This is such a creative study and so expertly executed. I really only have very minor recommendations for improvement. The lit review section would benefit from a paragraph that engages with research on women's "thin ideal" in US society. Susan Bordo's Unbearable Weight, Joan Brumberg 's The Body Project, and Lois Banner's American Beauty are my favorites for lit on that topic. The author should also link their findings about the feminization of health foods/ masculinization of high fat foods to the relevant bodies of research. Darwin's Omnivorous Masculinity and You are what you Drink contain fairly thorough lit reviews on this topic.

Other than that, there were a few longer sentences that read as awkward like line 303-306. But otherwise, this was exceptionally well done.

Author Response

Reviewer #2

This is such a creative study and so expertly executed. I really only have very minor recommendations for improvement.

Thank you for both your positive feedback and your constructive suggestions!

The lit review section would benefit from a paragraph that engages with research on women’s “thin ideal” in US society. Susan Bordo’s Unbearable Weight, Joan Brumberg ‘s The Body Project, and Lois Banner’s American Beauty are my favorites for lit on that topic.

Agreed! I’ve added a paragraph to the lit review with citations of the three resources you mention. (Unbearable Weight is one of my favorites; I’ve had a copy in my office for years!) I also added an additional reference to the Consuming Passions anthology of feminist reflections on eating disorders—which I highly recommend if this is a core area of interest in your own work.

The author should also link their findings about the feminization of health foods/ masculinization of high fat foods to the relevant bodies of research. Darwin’s Omnivorous Masculinity and You are what you Drink contain fairly thorough lit reviews on this topic.

Great suggestion. I should have thought of this in the first draft, as I had included one of Darwin’s publications on gender nonbinary experience already.

Other than that, there were a few longer sentences that read as awkward like line 303-306. But otherwise, this was exceptionally well done.

Thank you very much! I have done a general edit in the revised version to break up more complex or cumbersome sentences, including this one.